



# Incoming data quality control in high-resolution urban climate simulation: Hong Kong-Shenzhen area urban climate simulation as a case study using WRF/Noah LSM/SLUCM model (Version 3.7.1)

Zhiqiang Li[1], Bingcheng Wan[2], Yulun Zhou[3], and Hokit Wong[4]

[1]Department of Real Estate and Construction, The University of Hong Kong, Hong Kong SAR, 999077, China
[2]Glarun Technology Co., Ltd., Nanjing, 211100, China
[3]Department of Geography and Resource Management, The Chinese University of Hong Kong, Hong Kong SAR, 999077, China
[4]City University of Hong Kong, Hong Kong SAR, 999097, China

*Correspondence to*: Zhiqiang Li (paterlee@hku.hk)

Zhiqiang Li and Bingcheng Wan contributed equally to this work and should be considered co-first authors.

**Abstract.** Growing computational power in recent years enabled high-resolution urban climate simulations using limited-area models to flourish. This trend empowered us to deepen our understanding of urban-scale climatology with much finer spatial-temporal details. However, these high-resolution models would also be particularly sensitive to model uncertainties, especially in

urbanizing cities where natural surface texture is changed artificially into impervious surfaces with extreme rapidity, and these artificial changes always lead to dramatic changes in the land surface process. While models capturing detailed meteorological processes are being refined continuously, the input data quality has been the primary source of biases in modeling results but has received inadequate attention. To address this issue, we first examine the quality of the incoming static data in two cities in China, i.e., Shenzhen and Hong Kong SAR, provided by the WRF ARW model, a widely-applied state-of-the-art mesoscale numerical

weather simulation model. Shenzhen was going through an unprecedented urbanization process in the past thirty years, and Hong Kong SAR is another well-urbanized city. A significant proportion of the incoming data are found out-dated, which highlights the necessity of conducting incoming data quality control in the region of Shenzhen and Hong Kong SAR. Then, we proposed a sophisticated methodology to develop a high-resolution land surface dataset in this region. We conducted urban climate simulations in this region using both the developed land surface dataset and the original dataset utilizing the WRF ARW model coupled with

Noah LSM/SLUCM and evaluated the reliability of modeling results. The reliability of modeling results using the developed high-resolution urban land surface datasets is significantly improved compared to modeling results using the original land surface dataset in this region. This result demonstrates the necessity and effectiveness of the proposed methodology. Our results provide evidence on the effects of incoming land surface data quality on the accuracy of high-resolution urban climate simulations and emphasize the importance of the incoming data quality control.

**1 Introduction**

With the numerical weather prediction model applied in climate study increasingly in past decades (Warner, 2011), many scientists (such as Anthes, 1983; Keyser and Uccellini, 1987; Oreskes et al., 1994; Kain et al., 2008; Warner, 2011; Teutschbein and Seibert, 2012; Hong and Kanamitsu, 2014) were arguing the best modeling practices or provided suggestions related with modeling. Warner introduced the concept of quality assurance for improving modeling practices base on the summarization of wisdom in the previous

studies (Warner, 2011). In recent years, with rapidly developing computational capabilities, researchers become capable of applying high-resolution limited-area models to produce detailed meteorological scenarios. This capability empowers the studies of urban-scale climatology that require finer grid spacing. At the same time, this kind of high-resolution urban modeling also poses a significant challenge to the assurance of modeling quality. High-resolution urban climate simulation is sensitive to the quality of input of urban land surface data (Bruyère et al., 2014). An essential process that many pieces of research have been overlooked is





the incoming quality control, which guarantees the quality of input data for climate modeling. The incoming data quality control (IDQC), the procedure that ensures the accuracy of input data, is often ignored in modeling practices. In the manufacturing process, incoming quality control (IQC) is one of quality assurance activities because it is "a proactive upstream approach that controls and manages the upstream activities to prevent problems from arising" (World Meteorological Organization, 2014). By analogy with

the manufacturing process, IDQC of climate modeling resembles the incoming material quality control because it also is a proactive activity of controlling and managing the quality of the input data before the model-run rather than a "correction after problems occur" (World Meteorological Organization, 2014). Modeling practices using inaccurate input usually lead to significant modeling bias. However, sometimes, such modeling practices may still lead to seemingly-correct but scientifically misleading modeling results, which like window dressing to some extent. Therefore, a more sophisticated IDQC procedure is needed, which is a critical action

of quality assurance.

Urban climate modeling is notoriously sensitive to land surface characters (Chen, 2004; Sertel, 2010). The fallacious spatial distribution of the land surface data causes inaccurate values to be assigned to boundary layer variables. So the fallaciousness would be finally propagated to the modeling results, including temperature, wind fields, precipitation, and humidity (Chen, 2004). High-resolution models are more sensitive to incoming data quality for producing more refined details in finer spatial grids (Chen, 2004;

Sertel, 2010).

Despite the close association between the quality of land surface dataset and the quality of modeling results, IDQC has received inadequate attention in many existing modeling practices. Many urban climate systems provide a default and elemental input land surface dataset, the quality of which varies dramatically over space and time. For example, the WRF ARW model only provides detailed urban information for some big American cities. In most areas except these big cities, not only the recording date of the

default input land surface dataset for the WRF ARW model is far before 2010 but also the gird size of the one is far larger than 1 km. First, respective land surface datasets need to be developed if the study period is out of the temporal coverage of the default dataset. Second, while the quality of the default dataset in these big American cities is found acceptable, the quality of the same dataset in many other parts of the world can be quite problematic, especially in the rapidly urbanizing China. In the developing world, dramatic artificial interferences change natural surface texture into impervious surfaces with extreme rapidity. A compelling

example would be Shenzhen, China, a city that went through dramatic urbanization in the past thirty years.

In this study, a sensitivity test was conducted to examine the influence of urban land surface data accuracy on the urban climate modeling quality. Moreover, the article also proposed an explanation of why urban land surface accuracy affected urban climate modeling accuracy and stated the importance of the incoming quality control in urban climate modeling.

The remainder of the paper is organized as follows. Section 2 introduces the methodologies used in this paper. Section 3 presents

the comparison results between the default land surface dataset and the developed high-resolution dataset, and the comparison results between the corresponding modeling results. Section 4 sums up the discussions and concludes the paper.

## 2 Methodologies

### 2.1 Experimental design

In this study, we took the megacity's region in PRD: Hong Kong-Shenzhen area as the study area and delineated it in the modeling

as the inner-most domain (domain 4, Figure 2). We conduct two modeling experiments, respectively using the default land surface dataset and the developed high-resolution urban land surface dataset to examine if the IDQC improves urban climate modeling results. We compared not only the quality of the incoming data in the two urban climate simulation cases but also the quality of the modeling results to demonstrate the effects of the incoming data quality on high-resolution urban climate modeling. In this study, two comparative urban climate simulation cases were designed for evaluating the impacts of the refinement using the urban land

surface dataset on the quality of the modeled results. So, the modeled results from simulations using the WRF ARW/Noah LSM/SLUCM model with and without a refinement by the urban land surface dataset on the primary data would be compared. Case-



NCAR is a one-year climate simulation using the default land surface dataset provided by NCAR. Case-ULSD is a comparative experimental case using the developed high-resolution urban land surface dataset. Both cases are utilizing the same lateral boundary conditions in 2010 and the same model settings.

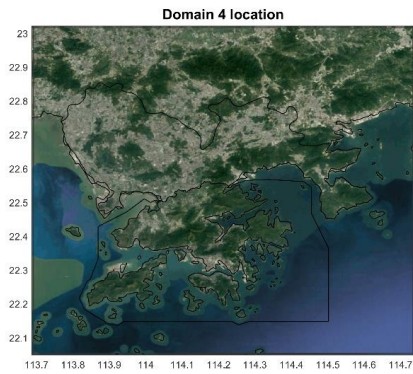

**Figure 1: The study area where was delineated in the modeling as the inner-most domain (domain 4), base map source: © Google Maps**

### 2.2 Evaluation methods for incoming data quality

We compared the original and refined land surface dataset by respectively analyzing its four major components – land cover, vegetation coverage, urban morphology, and anthropogenic heat fluxes - meteorological variables that are involved the most in

urban climate processes at intra-urban scale.

We also compared the simulation results using land surface data before and after the refinement to investigate the impact of incoming data quality on the accuracy and granularity of simulation results. Typically, the quality of urban climate modeled results should be evaluated by comparing the modeled near-surface variables with its corresponding observed ones (Li et al., 2019). We extracted and compared five near-surface meteorological variables – surface temperature and near-surface air temperature, wind speed,

precipitation, and relative humidity – respectively, along the spatial and temporal dimension. Moreover, these critical modeled near-surface variables are also compared with its corresponding observed ones by the statistic tools suggested by Li et al. (2019), which includes a temporal comparison of spatial variation (TCSV), Perkins skill score (PSS), and PDF of difference (PDFD). Furthermore, the near-surface variables of two experimental cases are compared spatially to analyze the difference between the two cases in the spatial dimension as well.

**2.3 Data**

The primary data includes the *Completed Dataset and the New Static Data Released With v3.7 of WRF Preprocessing System (WPS) Geographical Input Data* and the *2010 NCEP FNL (Final) Operational Global Analysis Dataset with 1-degree grid spatial resolution and the 6-hour temporal resolution*. Moreover, to demonstrate the effectiveness of the proposed methodology, we developed a high-resolution urban land surface dataset for Hong Kong-Shenzhen area in the year of 2010, which includes six kinds

of data, including land cover data, vegetation coverage data, urban morphology data, artificial impervious area data, and anthropogenic heat data, respectively describing different characters of the urban land surface. Furthermore, the *2010 MODIS/Aqua Land Surface Temperature and Emissivity (LST/E) product* and the *2010 near-surface metrological observation data in PRD* were used for modeled results' quality evaluation. Finally, the comparative meteorological variables are listed in Table 1.

**Table 1: Comparative meteorological variables.**

| Modeling results | Observation |
|---|---|
| Surface Skin Temperature | 2010 MODIS/Aqua Land Surface Temperature and Emissivity (LST/E) product [1] |





| 2-meters air temperature | 2010 PRD 2-Meters Air Temperature [2] |
| 10-meters wind at U direction | 2010 PRD 10-Meters Wind Speed [2] |
| 10-meters wind at V direction | |
| Accumulated total cumulus precipitation | 2010 PRD Precipitation [2] |
| Accumulated total grid-scale precipitation | |
| 2-meters relative humidity [2] | 2010 PRD Relative Humidity [2] |

Remarks: 1 - The data was provided by NASA EOSDIS Land Processes DAAC, USGS Earth Resources Observation and Science (EROS) Center. 2 - The data was provided by Meteorological Bureau of Shenzhen Municipality.

## 3 Results

### 3.1 Refinements in urban land surface data

5   Through the comparison between two cases, we found the default terrestrial input data missed many details in land cover (Figure 2), the vegetation coverage (Figure 3), urban morphology (Figure 4), and anthropogenic heat (Figure 5).

(a)                                                    (b)

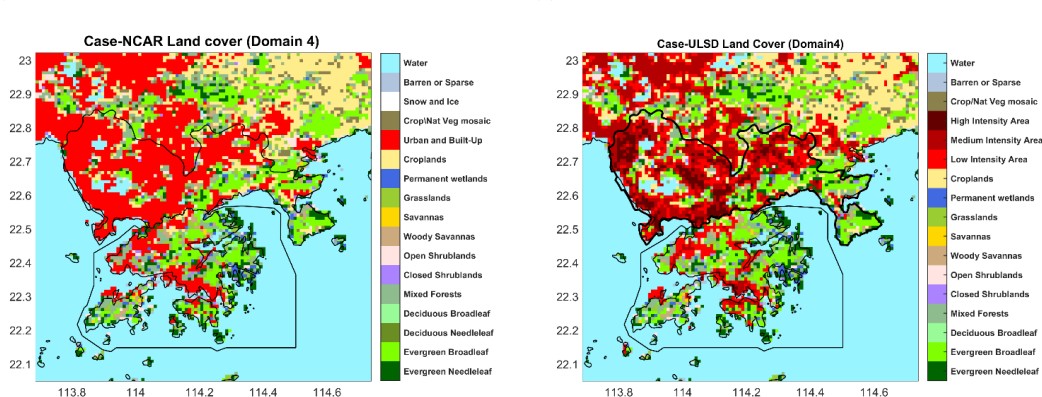

Figure 2: The land covers provided by NCAR originally (a), and the adjusted land covers based on the urban land surface dataset (b),
10   data source: Li et al., 2019 a and b.

(a)                                                    (b)

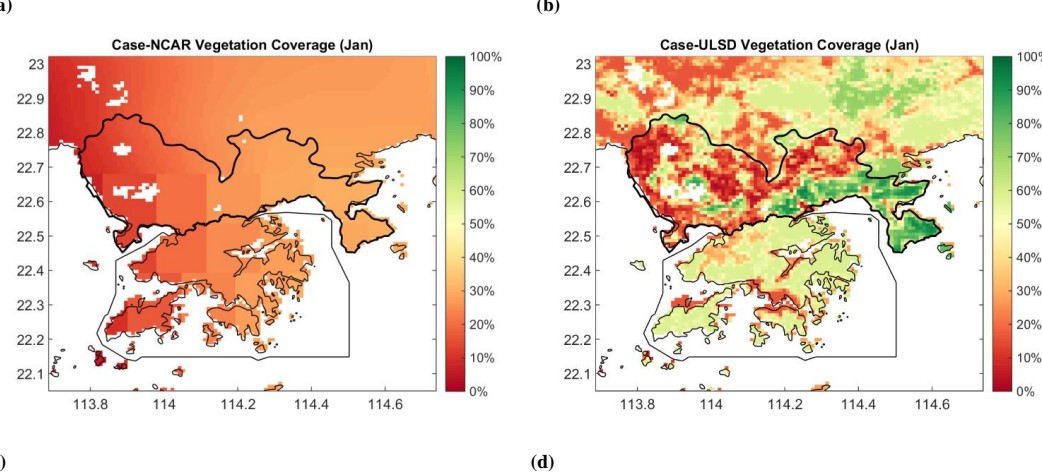

(c)                                                    (d)





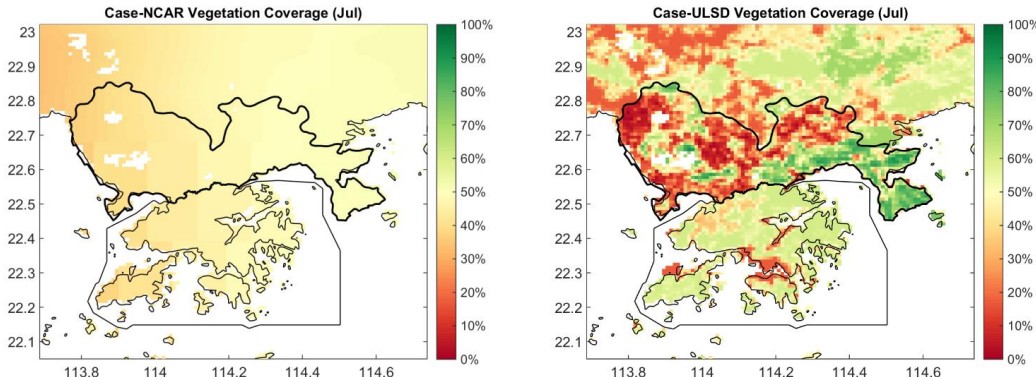

Figure 3: The original (left) and refined (right) vegetation coverage data product in January (upper, local winter) and July (lower, local summer), data source: Li et al., 2019 a and b.

(a)                                                          (b)

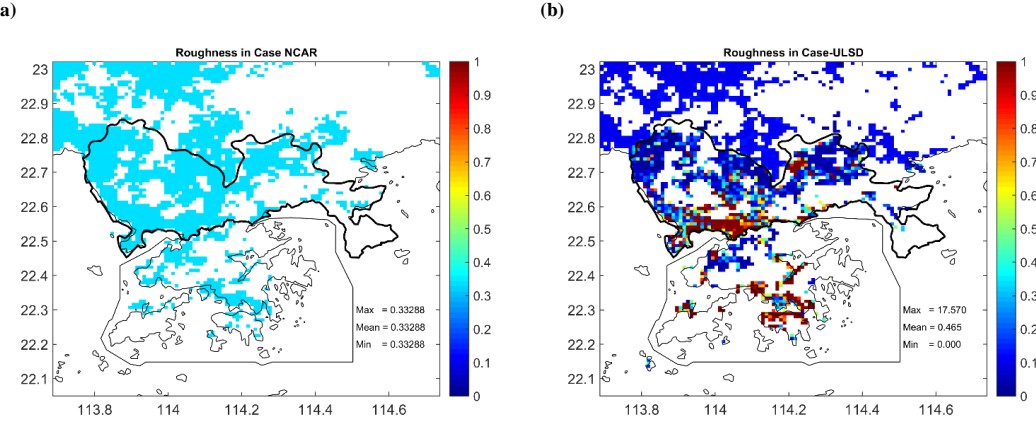

Figure 4 The number mean building height before (a) and after refinement (b), data source: Li et al., 2019 a and b. For other urban morphology indicators, please refer to Section S1 of Supplementary Material.

(a)                                                          (b)

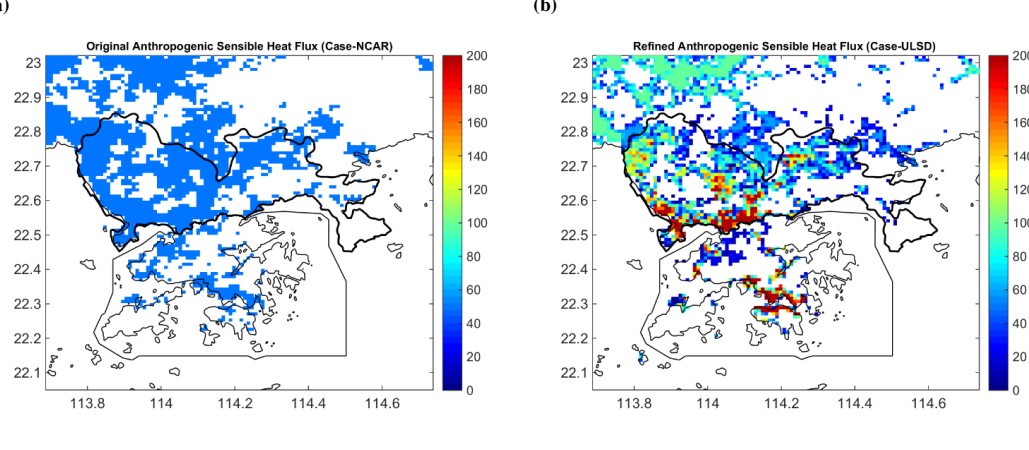

10   (c)                                                     (d)





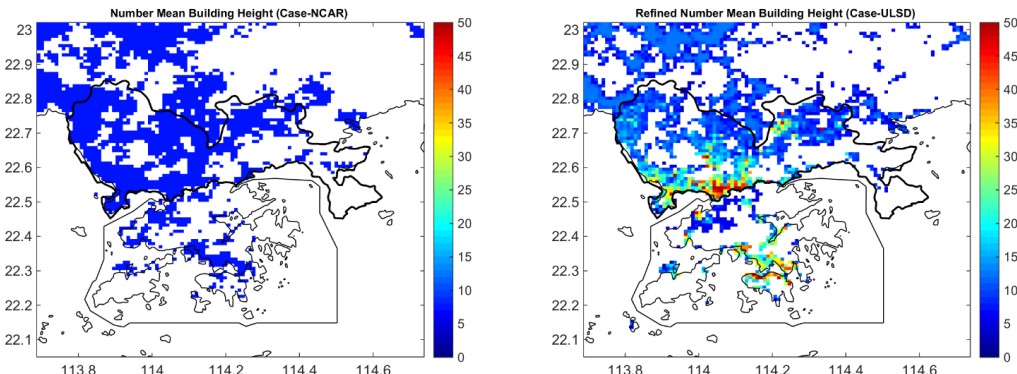

**Figure 5: Maps of the anthropogenic sensible (upper) and latent (lower) heat fluxes before (left) and after (right) refinement. , data source: Li et al., 2019 a and b.**

### 3.2 Quality of simulation results

With the TCSV figure, the temporal patterns of the simulation results using the land surface data before and after the refinement were examined against the temporal patterns in observations. By temporal patterns, we refer to both the diurnal and monthly variations of the selected meteorological variables. As evidenced by Figure 5, both simulation results using the original and refined land surface data reproduced the diurnal and monthly patterns as the ones of observation. Other meteorological variables of both cases, such as land surface temperature and relative humidity, precipitation, and wind speed, also have similar temporal variation

patterns as the corresponding observed ones.

(a)                                                                                      (b)

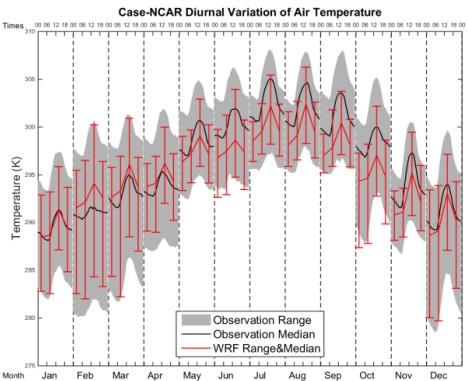 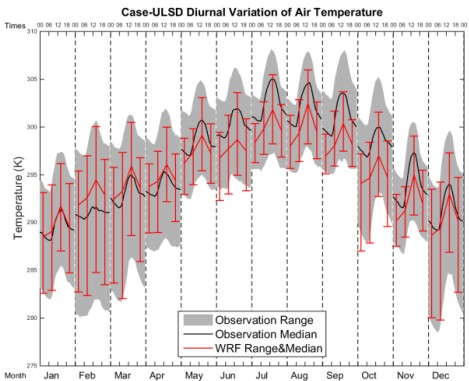

**Figure 6: The annual and diurnal variation of 2-meters air temperature of Case-NCAR (a) and Case-ULSD (b, data source: Li et al., 2019**

**a and b).**

Compared to Case-NCAR, the PSS annual mean values of Case-ULSD improved by 1.0%, 3.2%, and 5.5% in the 2-meters air temperature, surface temperature, and 10-meters wind speed, respectively. On the contrary, the PSS annual mean values of Case-ULSD deteriorated 5.6% and 2.7% in relative humidity and precipitation, respectively, than the ones of Case-NCAR. Figure 7 shows the PSS monthly variation of the 2-meters air temperature. Meanwhile, the PSS monthly variations of other meteorological variables

(surface temperature, and relative humidity, precipitation, and wind speed) are shown in Section S2 of Supplementary Material.

(a)                                                                                      (b)



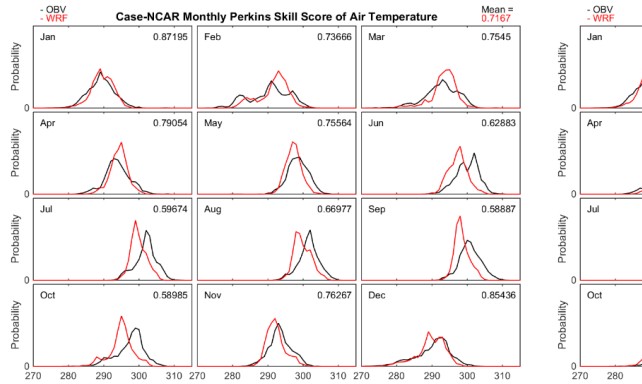
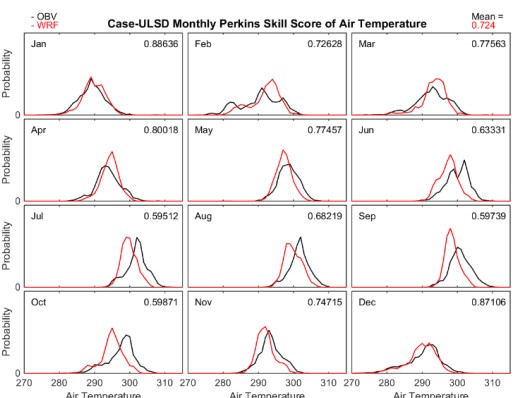

**Figure 7: PSS monthly variation of 2-meters air temperature of Case-NCAR (a) and Case-ULSD (b, data source: Li et al., 2019 a and b).**

Figure 8 shows the PDFD of 2-meters air temperature of two experimental cases. For the PDFD of surface temperature, relative humidity, precipitation, and 10-meters wind of two cases, please refer to Section S3 of Supplementary Material. Compared to Case-

5   NCAR, the annual mean values of the specified interval of the PDFD of Case-ULSD improved 2% in surface temperature and precipitation. Moreover, the annual mean values of the specified interval of the PDFD of Case-ULSD are the same as its corresponding ones of Case-NCAR in 2-meters air temperature and 10-meters wind speed. Furthermore, the annual mean values of the specified interval of the PDFD of Case-ULSD deceased 1% in relative humidity than the ones of Case-NCAR.

(a)                                                            (b)

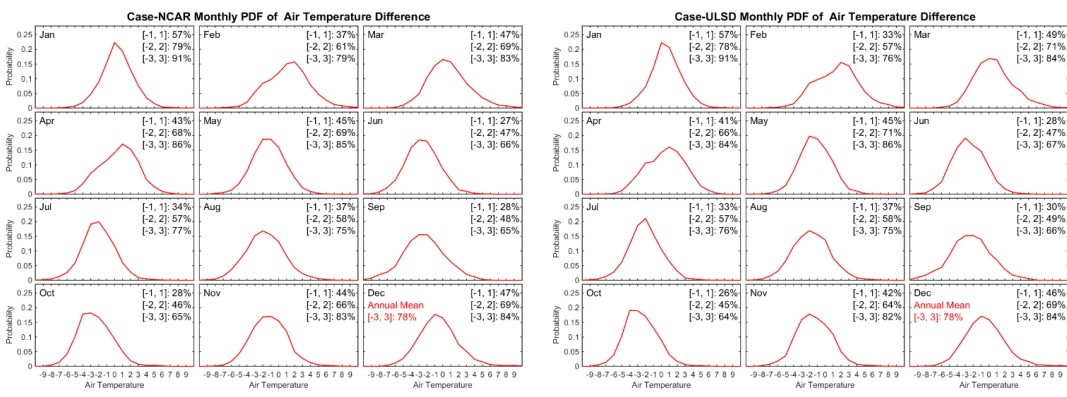

**Figure 8: PDFD of 2-meters air temperature of Case-NCAR (a) and Case-ULSD (b, data source: Li et al., 2019 a and b).**

To demonstrate the advantage of using the refined high-resolution land surface data, Figure 9 illustrates the spatial distributions of the annual mean values of the simulated 2-meters air temperature and surface temperature using the original and refined land surface data. For other meteorological variables (relative humidity, precipitation, and 10-meters wind) of two cases, please refer to Section

15   S4 of Supplementary Material. For all meteorological variables, the data refinement enabled more spatial details in the simulation results – could be increased or decreased in values. Such differences were especially significant for urban centers where massive and rapid urbanization has taken place. The urban-rural differences were strengthened, especially for temperatures and the wind speed. The air temperature and surface temperature in urban areas mostly increased with the refined datasets as a result of the increased urban area and newly included effects of urban morphology. On the contrary, wind speed and humidity mostly decreased

20   in urban areas.

(a)                                                            (b)



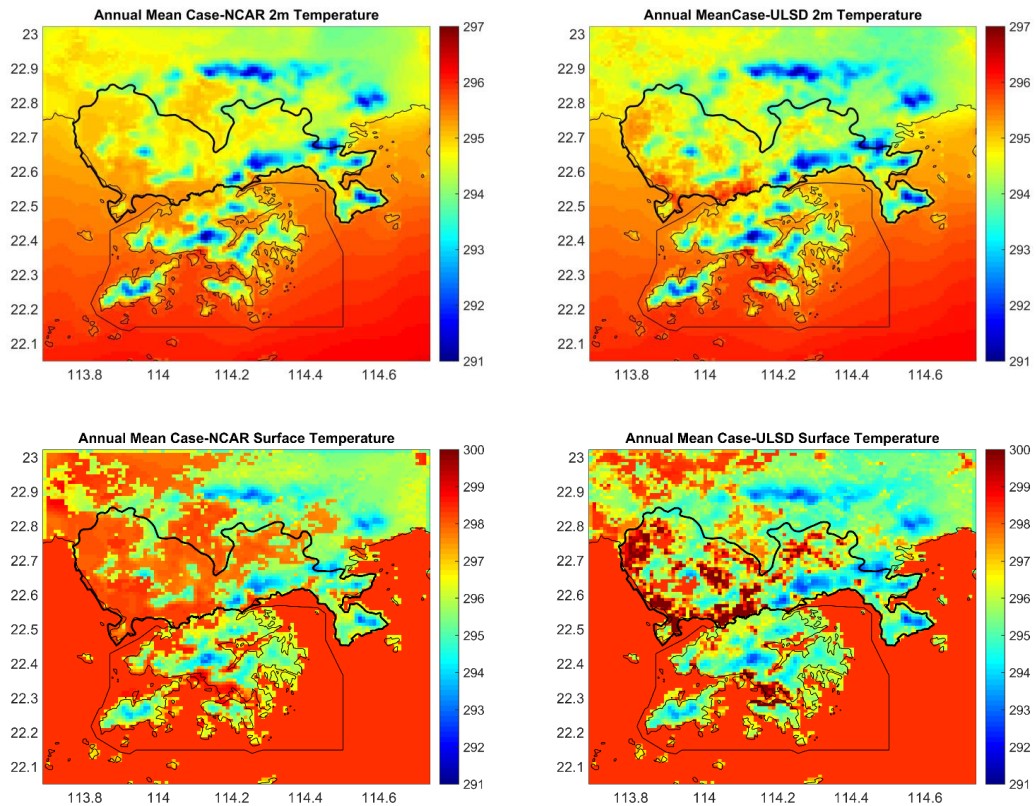

**Figure 9: Spatial distribution of 2-meters air temperature and surface temperature of Case-NCAR (a) and Case-ULSD (b), data source: Li et al., 2019 a and b.**

## 4 Discussions and conclusions

Undoubtedly, the high-quality land surface input data influence the modeling results that provide more distinct spatial details. Took a further step, we conducted a cause analysis of the physical mechanism of how the detail spatial features were transferred from urban land surface to urban atmosphere. The in-homogeneity of the urban land surface affects energy and mass redistribution in the atmosphere. Atmospheric models are composed of interacting components (Figure 10), which represent certain atmospheric physics principles (Dudhia, 2014). The WRF ARW model can be applied as a regional climate model, even though it was originated as a numerical weather prediction (NWP) model (Dudhia, 2014). By coupling it with Noah LSM/UCM model, the near-surface atmospheric physical processes of momentum, water vapor, and the heat exchange in an urban environment are represented in the WRF ARW model, thus enabling an improvement in the results of urban climate simulation (Tewari et al., 2007). The Noah LSM Model provides the different quantities to the WRF ARW model, such as the surface sensible heat flux, surface latent heat flux, upward longwave radiation, and upward shortwave radiation, based on the different textures of the surface. In an urban area, the anthropogenic heat release caused by human activities increases the surface sensible heat flux, and the urban morphology also has an impact on the radiation exchange and the roughness. The UCM model captures these urban land surface characteristics to improve the accuracy of the near-surface atmospheric physical processes.





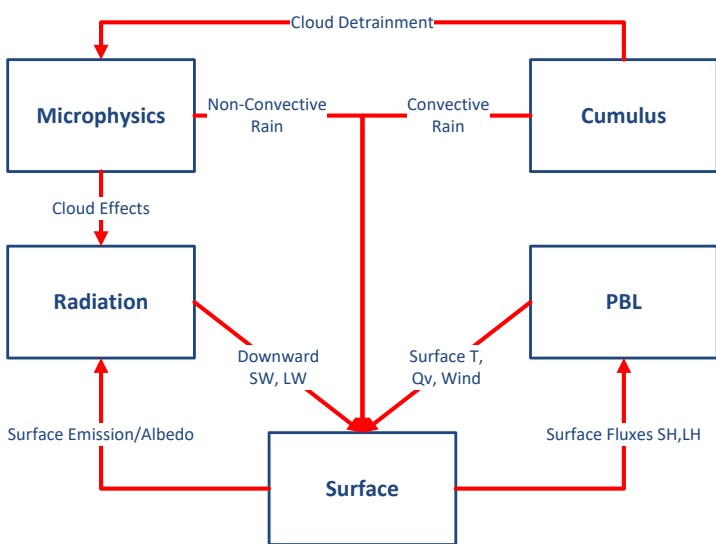

**Figure 10: Schematic of physics and their interactions within a typical NWP model (Dudhia, 2014).**

Moreover, when we looked inside the WRF ARW/Noah LSM/SLUCM Model, there are several physical modules in the model. These modules interact with each other to update the status variables iteratively to propagate the land surface attributes' detail spatial

5  features to the atmospheric variables (Figure 11). The detail spatial features in the land cover, vegetation coverage, urban morphology, and anthropogenic sensible and latent heat made effects on the 2-meters air temperature, relative humidity, precipitation and 10-meters wind speed by the intermediate variables (long and short wave radiations, surface temperature, surface sensible and latent heat).

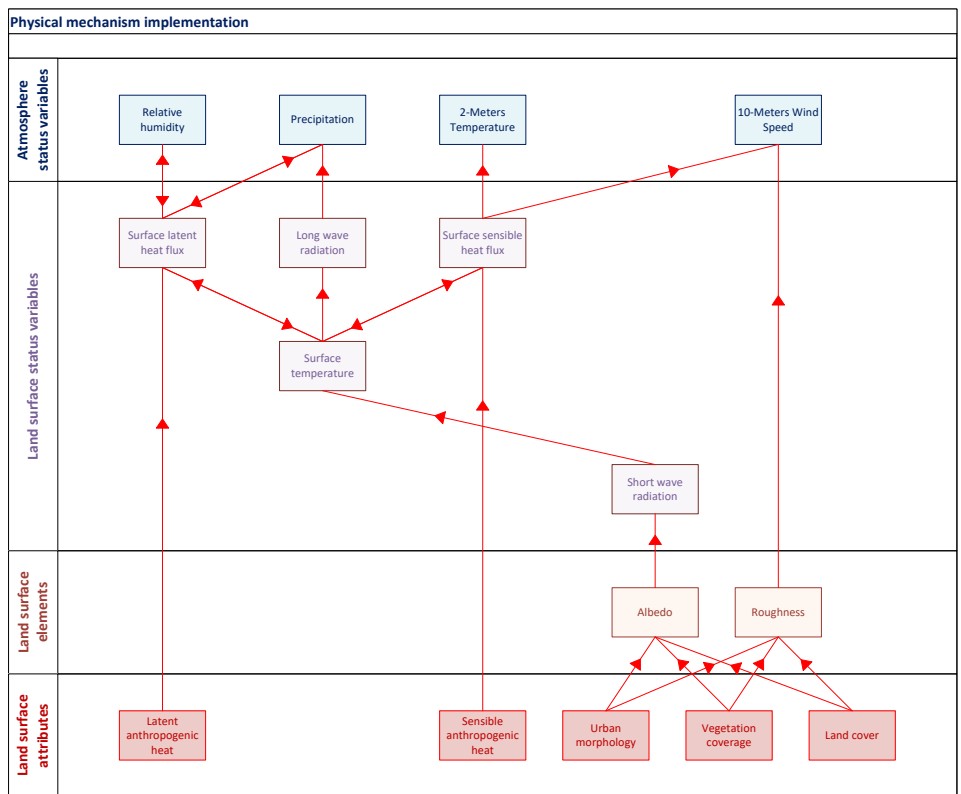



**Figure 11: The mechanism in propagating the detail spatial features from the land surface attributes to the atmospheric variables.**

For example, The 2-meters air temperature has a similar spatial pattern, like the ones in the albedo and sensible anthropogenic heat whatever in Case-NCAR and Case-ULSD. Moreover, the existence of this propagating mechanism can be seen from the transmission of the difference between two cases in the aforementioned intermediate variables. The spatial differences in land cover,

5  vegetation coverage between Case-NCAR and Case-ULSD caused the spatial difference in albedo (Figure 12a) between them. The differences in albedo and sensible anthropogenic heat (Figure 11b) between them led to the spatial differences in the sensible heat flux (Figure 12c) between them. In turn, these spatial differences led to the spatial differences in 2-meters temperature (Figure 12d).

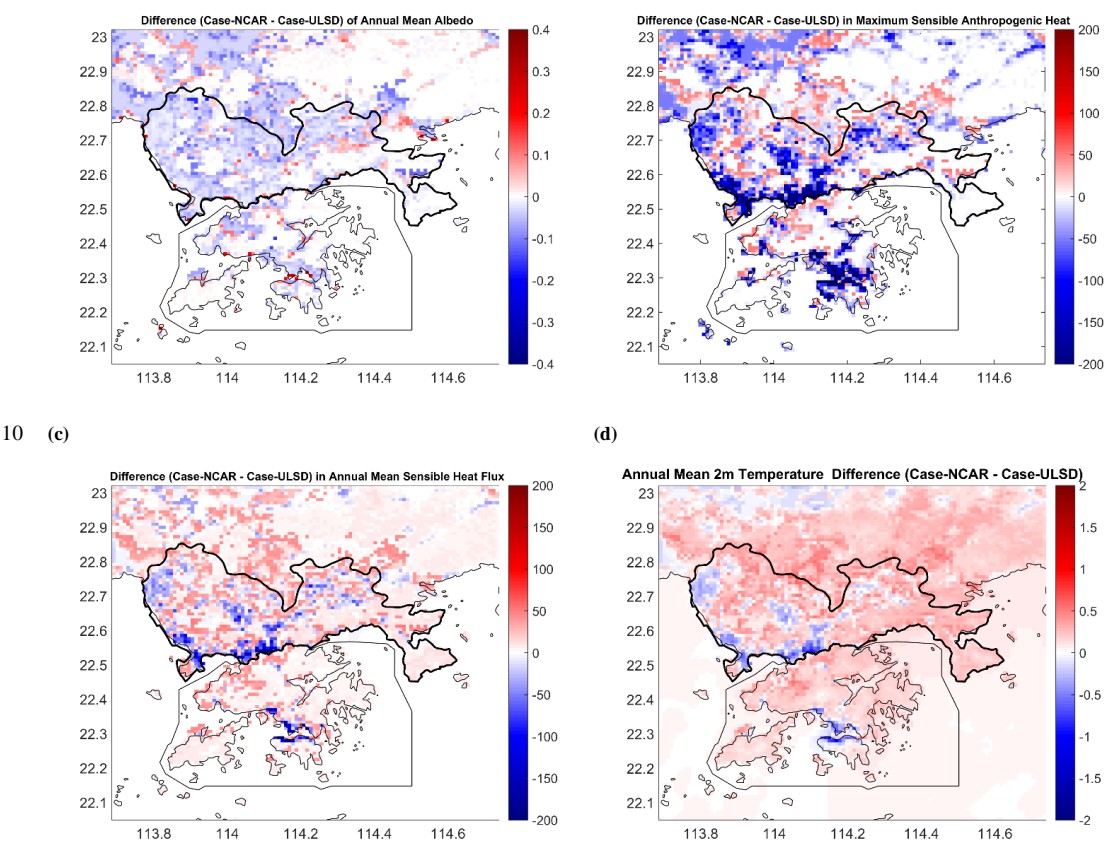

**Figure 12: Differences (Case-NCAR - Case-ULSD) in the albedo (a) annual mean sensible heat (a), the annual mean 2-meters air temperature (a).**

Distortion of modeling is nothing short of the problem in the climate modeler community. The spatial or temporal distortion in

15  climate modeling means that the result of the simulation is divergent with observation data spatially or temporally. Many scientists have addressed the problem since the 1980s and went on improving modeling practice in response to it (Warner, 2011). While improving the modeling practice, quality assurance should not be neglected in the climate modeling project (Warner, 2011), from an engineering project point of view. Quality assurance is "a part of quality management, but it is focused on providing confidence that quality requirements will be fulfilled" (ISO, 2005). However, quality assurance is overlooked due to the modeler has less

20  awareness of the sensitivity of the model (Warner, 2011). As a typical case of being careless, the IDQC, which makes sure the input data are accurate, is often ignored. Incoming quality control (IQC) is a familiar concept in production, which refers to the quality checking and evaluating of the incoming material. In the scenario of climate modeling, IQC is the quality control in the very first stage, which refers to the quality checking and evaluating of the incoming data before it will be delivered to the modeling system.





From our findings, the IDQC indeed improved the modeling results at the spatial dimension, creating substantially more spatial details in simulation results.

On the other hand, the modeled variables of both cases have the same temporal variation behaviors as its corresponding observed ones, irrespective of the 2-meters air temperature, surface temperature, relative humidity, precipitation, or 10-meters wind speed.

However, the values of PSS of 2-meters air temperature, surface temperature, and 10-meters wind speed in Case-ULSD are a little bit higher than those in Case-NCAR. The differences indicate that the modeling quality of Case-ULSD does not significantly improve when comparing to that of Case-NCAR. Worse still, the values of PSS of relative humidity and precipitation in Case-ULSD are lower than the ones in Case-NCAR, which means the modeling quality of Case-ULSD in these variables are worse than the ones of Case-NCAR. In the same manner, the annual mean values of the specified interval of the PDFD of Case-ULSD do not have a

significant improvement in the modeled 2-meters air temperature, surface temperature, precipitation, and 10-meters wind speed than the ones of Case-NCAR. It is even worse than the ones of Case-NCAR in the modeled relative humidity, which means the high-quality input data did not make a positive effect on the accuracy of the modeling results.

It is a valuable scientific question to probe the reason why the high quality land surface data does not improve the accuracy of the modeling results. The modeled humidity and precipitation are impacted majorly by two soil variables (soil water permeability and

initial relative soil moisture). We just used the default values of these two variables provided by NCAR in Case-ULSD. It would be a reason why the modeled humidity and precipitation never improved. Moreover, many physical parameterization processes involved in the physics components (Cumulus Parameterization, Microphysics, Radiation Planetary Boundary Layer, Surface layer, Land Surface Model, and Urban Canopy Model) of WRF ARW Noah LSM/SLUCM model. The default parameters of these processes were set suitable for the coarse resolution land surface data to ensure overall robustness in global uses. They accordingly

cannot guarantee that the adaptability for the fine resolution land surface data. It is the main reason to the question why the quality of land surface data did not transit the positive effects on the quality of modeling results, which sheds light on the direction in urban climate model development. The land surface data will become more and more refined in the future, and the parameterization schemes of the numerical model need to adapt these more refined data. Perhaps the parameterization schemes need to be further improved and consider more physical processes involved in the model.

Also, we only focused on the IDQC in the land surface data in this study. The soil attribute and initial condition should also be considered to conduct the IDQC for improving the quality of modeling results. Moreover, the quality of FNL data is critical to the accuracy of the lateral boundary conditions. The resolution of the currently used FNL data is relatively coarse. Accordingly, further follow-up work can be considered to improve the FNL data employing variational assimilation for improving the quality of urban climate modeling results.


*Code availability*.

The source codes of the WRF ARW modeling system package (WRF Model 3.7.1 and WRF Pre-Processing System (WPS) 3.7.1 ) are publicly available at http://www2.mmm.ucar.edu/wrf/users/download/get_source.html.

The configuration profile of the WRF ARW modeling system (namelist.wps and namelist.input), the changes in the WRF ARW

modeling system (the source codes for inputting the 2D anthropogenic sensible and latent heat), geo_data_refinement processing package and wrf_input_refinement processing package are available upon request from the corresponding author.

*Data availability*.

2010 NCEP FNL (Final) Operational Global Analysis Dataset is available at  https://rda.ucar.edu/datasets/ds083.2/.

40   Completed Dataset of WRF Preprocessing System (WPS) Geographical Input Data is publicly available at http://www2.mmm.ucar.edu/wrf/users/download/get_sources_wps_geog.html.



2010 PRD Observation Locations, 2010 PRD Urban Land Surface Dataset, 2010 PRD 2-meter Air Temperature, 2010 PRD 10-Meters Wind Speed, 2010 PRD Precipitation, and 2010 PRD Relative Humidity are available upon request from the corresponding author.

2010 MODIS/Aqua Land Surface Temperature and Emissivity (LST/E) product is publicly available at
https://lpdaac.usgs.gov/dataset_discovery/modis/modis_products_table/myd11a1_v006.

Modeling Variables for Model Evaluation (T2, TSK, U10, V10, RAINC, RAINNC, RH2, and SWDOWN) is available upon request from the corresponding author.

*Author contributions*. ZL, as the leading author, designed the model configuration, conducted the model run, conceived and designed
the experiment and the analysis, performed the experiment and the analysis, contributed data, developed the related software packages, and wrote the paper. BW contributed the ideas in the explanations of the physical mechanisms, designed the model configuration, collected the data, designed the analysis methods, and programmed some related software packages. YZ collected the data, performed the analysis, and critically revised the paper. HW supported the computer systems and checked the paper for language errors.


*Competing interests*. The authors declare that they have no conflict of interest.

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
