# Peer review of "Incoming data quality control in high-resolution urban climate simulation: Hong Kong-Shenzhen area urban climate simulation as a case study using WRF/Noah LSM/SLUCM model (Version 3.7.1)"

_Geoscientific Model Development, 2020_

## Referee Comment (RC1) · Anonymous Referee #1 · 9 May 2020

The paper addresses the importance of incoming data quality control and presents a robust method for evaluating its impact on climate simulation results. The authors find that the high-quality land surface input data would provide more distinct spatial details in the modeling results, but would not bring significant improvement to weather prediction. This finding is a very interesting point, and is critical to the model development. The study is valuable to be published in a high impact journal.

[Figure]

Major comments and questions:

(1) Please provide more explanation on TCSV, PSS and PDF results. For example, what is the indication of 'the PSS annual mean values of Case-ULSD deteriorated 2.7% in precipitation' while 'the annual mean values of the specified interval of the PDFD of Case-ULSD improved 2% in precipitation'?

(2) The surface temperature seems very important in the modelling as shown in the Figure 11. Could you provide more explanation of the 'surface temperature'? In addition, is the 'surface skin temperature' in the MODIS/Aqua product the same as the 'surface temperature' from the WRF model? By the way, I really like your Figure 11, a very good presentation to show the interplay among different factors.

(3) Please add TCSV figures of land surface temperature, relative humidity, precipitation and wind speed in the supplementary material.

(4) Please provide more information for the model setting (e.g. model version, domain boundary) and data sources of input data.

(5) Based on your findings, what are your suggestions for modeling development, e.g. balancing incoming data quality and parameterization schemes for a better weather prediction?

Corrections: (1) Figure 4 and Figure 5: the plots and figure captions are not consistent. (2) In the first paragraph of section 3.2, it should be 'Figure 6' instead of 'Figure 5'.

---

## Referee Comment (RC2) · Anonymous Referee #2 · 6 Jul 2020

In this work, the authors examined the influence of urban land surface data accuracy on the urban climate modeling quality. They compared the modeled results from simulations using the WRF ARW/Noah LSM/SLUCM model with and without a refinement by the urban land surface dataset. They clearly showed the high-quality land surface input data influence the modeling results that provide more distinct spatial details. They also proposed some explanation of how urban land surface data accuracy affected urban climate modeling accuracy. The paper is well written. I have given my comments
below. Mostly minor.

Abstract, "The reliability of modeling results using the developed high resolution urban land surface datasets is significantly improved compared to modeling results using the original land surface dataset in this region." I agree that the modeling results provide more distinct spatial details. Please elaborate what are the significant improvements in the reliability of the modeling results. This point is not clear in the abstract and main text.

Page 6, line 7, "As evidenced by Figure 5, both simulation results using the original and refined land surface data reproduced the diurnal and monthly patterns as the ones of observation." Would it refer to Figure 6?

There are questions about the improvements in the model simulations

Page 6, Line 16, "Compared to Case-NCAR, the PSS annual mean values of Case-ULSD improved by 1.0%, 3.2%, and 5.5% in the 2-meters air temperature, surface temperature, and 10-meters wind speed, respectively. On the contrary, the PSS annual mean values of CaseULSD deteriorated 5.6% and 2.7% in relative humidity and precipitation, respectively, than the ones of Case-NCAR."

Page 7, line 4, "Compared to Case5 NCAR, the annual mean values of the specified interval of the PDFD of Case-ULSD improved 2% in surface temperature and precipitation."

For the annual mean values of these factors, could the authors further elaborate what are the improvements (e.g. reliability) in their modeling results?

Page 11, "From our findings, the IDQC indeed improved the modeling results at the spatial dimension, creating substantially more spatial details in simulation results." I agree this work provide more spatial details in simulation results. However, did the authors compare their spatial results to the measurements?

---

## Author Comment (AC1) · 6 Aug 2020

Dear Reviewer,

We appreciate you for spending time to review our paper and providing some valuable comments. It is your valuable and insightful comments that led to possible improvements in the current version. We have carefully considered the comments and tried our best efforts to address every one of them. However, some revisions still cannot

meet your high standards. The authors welcome further constructive comments, if any. We provided the point-to-point response first and will provide the updated version of the paper after proofreading complete. Below we provide the point-by-point responses. All modifications in the manuscript have been highlighted in red.

Sincerely,

Li, Zhiqiang

Ph.D. in Earth System and Geo-information Science

Department of Real Estate and Construction | HKU Faculty of Architecture

Cell: +852 60608137

Email: paterlee@hku.hk

[General Comment] The paper addresses the importance of incoming data quality control and presents a robust method for evaluating its impact on climate simulation results. The authors find that the high-quality land surface input data would provide more distinct spatial details in the modeling results but would not bring significant improvement to weather prediction. This finding is a very interesting point and is critical to the model development. The study is valuable to be published in a high impact journal.

Response: Thank you very much for your valuable comments. We are also happy that you agree with our points of view on climate model evaluation.

[Comment 1] Please provide more explanation on TCSV, PSS and PDF results. For example, what is the indication of 'the PSS annual mean values of Case-ULSD deteriorated 2.7% in precipitation' while 'the annual mean values of the specified interval of the PDFD of Case-ULSD improved 2% in precipitation'?

Response: Thank you very much for the comments. Indeed, we provided detailed explanations of the evaluation methods (TCSV, PSS, and PDF) in its companion paper - Li, Z., Zhou, Y., Wan, B., Chung, H., Huang, B., & Liu, B. (2019). Model evaluation

of high-resolution urban climate simulations: using the WRF/Noah LSM/SLUCM model (Version 3.7. 1) as a case study. Geoscientific Model Development, 12(11), 4571-4584. Moreover, we added a note in Subsection 2.2 [Pg3, Ln17-19].

[Comment 2] The surface temperature seems very important in the modelling as shown in the Figure 11. Could you provide more explanation of the 'surface temperature'? In addition, is the 'surface skin temperature' in the MODIS/Aqua product the same as the 'surface temperature' from the WRF model? By the way, I really like your Figure 11, a very good presentation to show the interplay among different factors.

Response: Thank you very much for the comments. In a physical meaning, the land surface temperature in MODIS/Aqua product and the surface skin temperature in WRF simulation are two different concepts. The land surface temperature is the grid-mean brightness temperature of the earth, which is calculated base on the blackbody radiation theory. The surface skin temperature is a land's state variable in WRF, which is adjusted iteratively in each calculation base on the balance of the radiation, the sensible heat flux, the latent heat flux, and the soil heat conduction flux. The value of the surface skin temperature may diff a bit from one of the land surface temperatures in the same grid. However, these two variables are highly correlated. Therefore, we use the land surface temperature to evaluate the quality of the surface skin temperature in the WRF simulation. We added a technical note regarding the surface skin temperature and the land surface temperature, [Pg3, Ln29 – Pg4, Ln4]. Moreover, we also corrected a mistake (Surface temperature is incorrect. The correct on is Surface skin temperature) in Figure 11.

[Comment 3] Please add TCSV figures of land surface temperature, relative humidity, precipitation and wind speed in the supplementary material.

Response: Revised accordingly: added Section S5 into the Supplementary material.

[Comment 4] Please provide more information for the model setting (e.g. model version, domain boundary) and data sources of input data.

Response: Thank you very much for the comments. We added Section S6 into the Supplementary material for providing more information on model setting. We already provided the information on data sources of input data in Subsection 2.3.

[Comment 5] Based on your findings, what are your suggestions for modeling development, e.g. balancing incoming data quality and parameterization schemes for a better weather prediction?

Response: It is a good question. Admittedly, the incoming data quality contributes to improving the quality of modeling results. The sensitivity of land surface processes to input land surface data are also crucial for improving the quality of modeling results. For further steps in the atmospheric model development, we suggested improving the sensitivity of the urban land surface model to the input land surface data. We added a suggestion for modeling development in Section 4.

[Comment 6] Corrections: (1) Figure 4 and Figure 5: the plots and figure captions are not consistent. (2) In the first paragraph of section 3.2, it should be 'Figure 6' instead of 'Figure 5'.

Response: Thank you very for the reminder. (1) Revised accordingly: Figure 4 [Pg5, Ln4], and Figure 5 [Pg6, Ln1]. (2) Revised accordingly, [Pg6, Ln8].

---

## Author Comment (AC2) · 6 Aug 2020

Dear Reviewer,

We appreciate you for spending time to review our paper and providing some valuable comments. It is your valuable and insightful comments that led to possible improvements in the current version. We have carefully considered the comments and tried our best efforts to address every one of them. However, some revisions may still not

meet your high standards. The authors welcome further constructive comments if any. We provided the point-to-point response first and will provide the updated version of the paper after proofreading complete. Below we provide the point-by-point responses. All modifications in the manuscript have been highlighted in red.

Sincerely,

Li, Zhiqiang

Ph.D. in Earth System and Geo-information Science

Department of Real Estate and Construction | HKU Faculty of Architecture

Cell: +852 60608137

Email: paterlee@hku.hk

[General Comment] In this work, the authors examined the influence of urban land surface data accuracy on the urban climate modeling quality. They compared the modeled results from simulations using the WRF ARW/Noah LSM/SLUCM model with and without a refinement by the urban land surface dataset. They clearly showed the high-quality land surface input data influence the modeling results that provide more distinct spatial details. They also proposed some explanation of how urban land surface data accuracy affected urban climate modeling accuracy. The paper is well written. I have given my comments below. Mostly minor.

Response: Thank you very much for your valuable comments. We are delighted that you agree with the views of this paper.

[Comment 1] Abstract, "The reliability of modeling results using the developed high resolution urban land surface datasets is significantly improved compared to modeling results using the original land surface dataset in this region." I agree that the modeling results provide more distinct spatial details. Please elaborate what are the significant improvements in the reliability of the modeling results. This point is not clear in the

abstract and main text.

Response: Thank you very much for the comments. We compared the model result with station observation and MODIS land surface temperature. We use the temporal comparison of spatial variation (TCSV), Perkins skill score (PSS), and PDF of difference (PDFD) to evaluate the model results. Most of the results show improvements in the Case-USLD. Indeed, the most significant improvement in modeling results by the incoming data quality control is that the model produced more distinct spatial details in the fine grids. Actually, urban climate modeling is a meteorological downscaling application that is employed to produce the fine-scale spatial and temporal details from the coarse resolution's meteorological data (Hong et al., 2014). It is, therefore, the critical indicator for the urban climate to precisely construct the fine-scale details at their utmost in the interested area (Lo et al., 2008). In this study, we conducted two meteorological downscaling cases by the dynamical limited area model with the same lateral boundary condition of coarse-resolution data and two different land surface data to compare which case constructs more fine details in the interested area. From the dynamical meteorological downscaling point of view, Case-USLD has a significant improvement in the performance of modeling results than the Case-NCAR. We added the content in the abstract and Section 4 to emphasize the points aforementioned [Pg13, Ln11-21] and changed the wording [Pg1, Ln25].

[Comment 2] Page 6, line 7, "As evidenced by Figure 5, both simulation results using the original and refined land surface data reproduced the diurnal and monthly patterns as the ones of observation." Would it refer to Figure 6?

Response: Thank you very much for reminding us. Revised accordingly, [Pg6, Ln8].

[Comment 3] There are questions about the improvements in the model simulations. (1) Page 6, Line 16, "Compared to Case-NCAR, the PSS annual mean values of Case-ULSD improved by 1.0%, 3.2%, and 5.5% in the 2-meters air temperature, surface temperature, and 10-meters wind speed, respectively. On the contrary, the PSS annual mean values of CaseULSD deteriorated 5.6% and 2.7% in relative humidity and precipitation, respectively, than the ones of Case-NCAR." (2) Page 7, line 4, "Compared to Case5 NCAR, the annual mean values of the specified interval of the PDFD of Case-ULSD improved 2% in surface temperature and precipitation." For the annual mean values of these factors, could the authors further elaborate what are the improvements (e.g. reliability) in their modeling results?

Response: Thank you very much for the comments. In this study, it is another interesting finding that the high-quality land surface did not make a positive effect on the modeling results. We also conducted a discussion for this finding in Section 4 [Pg12, Ln17 – Pg13, Ln10]. For responding to this question, we just changed the wording (replace "reliability" with "performance").

[Comment 4] Page 11, "From our findings, the IDQC indeed improved the modeling results at the spatial dimension, creating substantially more spatial details in simulation results." I agree this work provide more spatial details in simulation results. However, did the authors compare their spatial results to the measurements?

Response: Thank you very much for the comment. Yes, we spatially compared the modeling results of two cases with the observation by the PSS and PDFD, which include the wind speed, air temperature, relative humidity, precipitation, and surface skin temperature.

―――――――――――――――――――

---

## Author Comment (AC3) · 7 Sep 2020

Response to the Topical Editor [Cover letter] Dear Dr David Topping, We appreciate you for spending time to review our paper and providing some valuable comments. It is your valuable and insightful comments that led to possible improvements in the current version. We have carefully considered the comments and tried our best efforts to address every one of them. We provided the point-to-point response first and will provide

the updated version of the paper after proofreading complete. Below we provide the point-by-point responses. All modifications in the manuscript have been highlighted in red.

Sincerely,

Li, Zhiqiang Ph.D. in Earth System and Geo-information Science Department of Real Estate and Construction | HKU Faculty of Architecture Cell: +852 60608137 Email: paterlee@hku.hk

[Comment 1] From the code section, it now looks great...although the manuscript you uploaded to GMD still has the old 'contact the author' section?: The configuration profile of the WRF ARW modeling system (namelist.wps and namelist.input), the changes in the WRF ARW modeling system (the source codes for inputting the 2D anthropogenic sensible and latent heat), geo_data_refinement processing package and wrf_input_refinement processing package are available upon request from the corresponding author. Response: Thank you very much for reminding us. Revised accordingly, [Pg13, Ln28].

[Comment 2] You also request people contact you for some data rather than Zenodo or another archive. Is it possible to also put this on an archive? Response: Thank you very much for reminding us. We uploaded all data to Zenodo and revised the manuscript, [Pg13, Ln34; Pg 13, Ln40].

---

## Author Response (AR2)

**Response to the Topical Editor**

**[Cover letter]**

Dear Dr David Topping,

We appreciate you for spending time to review our paper and providing some valuable comments. It is your valuable and insightful comments that led to possible improvements in the current version. We have carefully considered the comments and tried our best efforts to address every one of them. We provided the point-to-point response first and will provide the updated version of the paper after proofreading complete.

Below we provide the point-by-point responses. All modifications in the manuscript have been highlighted in red.

Sincerely,

Li, Zhiqiang

Ph.D. in Earth System and Geo-information Science

Department of Real Estate and Construction | HKU Faculty of Architecture

Cell: +852 60608137

Email: paterlee@hku.hk

**[Comment 1]** From the code section, it now looks great...although the manuscript you uploaded to GMD still has the old 'contact the author' section?: The configuration profile of the WRF ARW modeling system (namelist.wps and namelist.input), the changes in the WRF ARW modeling system (the source codes for inputting the 2D anthropogenic sensible and latent heat), geo_data_refinement processing package and wrf_input_refinement processing package are available upon request from the corresponding author.

**Response:** Thank you very much for reminding us. Revised accordingly, [Pg13, Ln28].

**[Comment 2]** You also request people contact you for some data rather than Zenodo or another archive. Is it possible to also put this on an archive?

**Response:** Thank you very much for reminding us. We uploaded all data to Zenodo and revised the manuscript, [Pg13, Ln34; Pg 13, Ln40].

[revised manuscript text omitted]

**(a)**

**(b)**

[Figure]

[Figure]

[Figure]

[Figure]

[Figure]

[Figure]

[Figure]

[Figure]

[Figure]

[Figure]

[Figure]

[Figure]

[Figure]

[Figure]

[Figure]

[Figure]

[Figure]

[Figure]

[Figure]

[Figure]

[Figure]

[Figure]

[Figure]

[Figure]

**Figure S6: TCSV of relative humidity, precipitation, and 10-meters wind of Case-NCAR (a) and Case-ULSD (b, data source: Li et al., 2019 b) at 2:00, 8:00, 14:00, and 20:00.**

**S6 Model setting**

5    The model settings of both comparative urban climate simulation cases are the same as that of the companion study (Li et al., 2019 b). We set four telescoping nested domains with a center at 22°39'30" N, 114°11'30" as the horizontal domain configuration, and a set of eta level with 51 members for each horizontal domain as the vertical grid spacing configuration. Moreover, we configured the physics components with the schemes in Table S1.

**Table S1: The configuration of the physics components (Li et al., 2019 b).**

| Component | Scheme |
|---|---|
| Cumulus | New Simplified Arakawa-Schubert |
| Microphysics | WDM5 |
| Radiation | RRTMG |
| Planetary Boundary Layer | Bougeault–Lacarrere |
| Surface Layer | Revised MM5 |
| Land Surface Model | Noah LSM |
| Urban Canopy Model | Single-layer |

---

## Author Response (AR3)

**Response to the Topical Editor**

**[Cover letter]**

Dear Dr David Topping,

We appreciate you for spending time to review our paper. We also were very happy to know that our manuscript would be ready to publish. We received your commnet just before our long holiday. It is a great holiday gift to us. Thank very much again.

Below we provide the point-by-point responses. All modifications in the manuscript have been highlighted in red.

Sincerely,

Li, Zhiqiang

Ph.D. in Earth System and Geo-information Science

Department of Real Estate and Construction | HKU Faculty of Architecture

Cell: +852 60608137

Email: paterlee@hku.hk

**[Comments]** Dear authors. Thank you for taking the time to make sure all data is available. there is one slight error I have noticed in the manuscript. You have a sentence that is unfinished: 2010 PRD Observation Locations, 2010 PRD 2-meter Air Temperature, 2010 PRD 10-Meters Wind Speed, 2010 PRD Precipitation, 35and 2010 PRD Relative Humidity are available at . Please add the URL but we are then ready to publish :)

Thanks!

**Response:** Thank you very much for reminding us. Revised accordingly, [Pg13, Ln36].

[revised manuscript text omitted]